# Pathogens Detected in 205 German Farms with Porcine Neonatal Diarrhea in 2017

**DOI:** 10.3390/vetsci9020044

**Published:** 2022-01-25

**Authors:** Nicolas Mertens, Tobias Theuß, Monika Köchling, Karen Dohmann, Kathrin Lillie-Jaschniski

**Affiliations:** 1Ceva Tiergesundheit, Kanzlerstraße 4, 40472 Düsseldorf, Germany; nicolas.mertens@ceva.com (N.M.); monika.koechling@ceva.com (M.K.); 2Ceva Innovation Center, Am Pharmapark, 06861 Dessau-Roßlau, Germany; tobias.theuss@ceva.com; 3IVD GmbH, Albert-Einstein-Straße 5, 30926 Seelze-Letter, Germany; dohmann@ivd-gmbh.de

**Keywords:** neonatal diarrhea, fecal samples, Germany, *C. perfringens*, *E. coli*, Rotavirus, *C. difficile*

## Abstract

Neonatal diarrhea (ND) is still a frequently observed problem in modern industrial pig production. ND is predominantly caused by bacterial and viral pathogens. The objective of this study was to give an overview of different pathogens involved in ND in Germany. In 2017, a total number of 555 litters from 205 German pig farms with clinical ND were sampled with pooled fecal samples. All samples were analyzed regarding bacterial pathogens by culture and viral pathogens by polymerase chain reaction (PCR). Isolated strains of *Clostridium* (*C.*) *perfringens*, *Escherichia* (*E.*) *coli*, and *C. difficile* were further characterized by molecular techniques (e.g., PCR). There were 200 litters (36%), out of 555 sampled litters of 205 farms, which were positive for at least one, while most of them were positive for two or more pathogens. Toxin-producing *C. perfringens* type A could be detected in 122 farms (59.2%), *C. difficile* in 116 (56.1%), pathogenic *E. coli* in 79 (38.6%), and Rotavirus type A in 72 (35%). Among *E. coli* isolates, enterotoxigenic (8.8%) (F4 fimbriae positive (60.0%)) and necrotoxigenic *E. coli* (5.3%) were the most frequently detected pathotypes. In conclusion, in most of the farms with porcine ND it turned out to be a disease mainly caused by multiple pathogens, predominantly *C. perfringens* type A, pathogenic *E. coli*, and Rotavirus type A. Nevertheless, *C. difficile* and necrotoxigenic *E. coli* might be emerging pathogens in ND.

## 1. Introduction

Porcine neonatal diarrhea (ND) is still one of the major diseases in the modern swine-producing industry. Depending on the pathogens involved, ND can cause high morbidity and mortality and, thus, often leads to increased economic losses [1]. In the first days of life, numerous bacterial and viral agents can cause ND either as a primary or secondary pathogen. From the second week of life onwards, intestinal disease also can be caused by parasites, manly *Cystoisospora suis* [2]. Recent studies have shown that pathogenic *Escherichia* (*E.*) *coli*, such as enterotoxigenic *E. coli* or *Clostridium* (*C.*) *perfringens* type A, and Rotavirus type A are the most frequently detected pathogens in cases of porcine ND [3,4] *C. perfringens* Type C, which causes highly fatal, necrohemorrhagic enteritis, lately seem to play a minor role as they are less frequently involved in cases of ND in recently published studies [3,4]. Besides, these well-described pathogens and the role and relevance of other bacteria detected in cases of porcine ND is not clarified yet. Among these, *C. difficile* and necrotoxigenic *E. coli* (NTEC) are worth mentioning. In human medicine, *C. difficile* is the etiologic agent of pseudomembranous colitis, a severe, fatal disease that occurs in adults associated with antimicrobial therapy. Additionally, it might be a relevant foodborne pathogen [5]. Porcine ND caused by *C. difficile* can occur in piglets between the first and seventh days of life, sometimes associated with sudden death [6]. There is only little information about the current prevalence of *C. difficile* in cases of porcine ND, and its role in porcine enteric diseases is not completely understood [7]. Regarding NTEC, the situation is comparable. While it is proven that NTEC isolates can cause ND in humans and ruminants [8,9], it is unknown whether those strains might contribute to porcine ND. Furthermore, only one report describes the prevalence of NTEC in pig isolates [3]. Viral causes of ND are mainly Rotavirus type A–E and two different types of Coronaviruses, the Transmissible Gastroenteritis Virus (TGEV) and the Porcine Epidemic Diarrhea Virus (PEDV). Rotavirus type A (RVA) has been considered as the main viral infectious agent of ND until now, but types B and C have been more frequently detected in recent years [10]. In most cases, it is almost impossible to state a reliable etiological diagnosis only based on clinical signs. In affected herds, co-infections with other enteric pathogens and non-infectious causes frequently impair clinical signs. Therefore, case-based and complex laboratory examination is necessary [2], together with pathoanatomic investigation, to verify whether a certain isolated pathogen has indeed caused a specific lesion. However, in many cases the number of animals sent to diagnostic investigation is insufficient to obtain a result representative for the farm [11]. The aim of the present study was to give an overview on the presence of bacterial and viral pathogens in fecal samples from piglets suffering from ND in Germany in 2017.

## 2. Materials and Methods

In total, 205 sow farms located in Germany, with reported recent history of ND, were sampled to get an overview of enteric pathogens involved in cases of ND in Germany. From January until December 2017, German swine veterinarians submitted fecal samples of three affected litters for further investigations. In Figure 1, the location of the farms is shown on a map of Germany. From each affected litter, the feces of five piglets showing diarrhea were pooled in a plastic tube. Only feces directly at defecation had to be used, no samples were collected from the floor. Included litters had to be untreated (no antibiotic pretreatment) and between 1 and 7 days of age. If one sample out of a submission was positive for a pathogen, the whole farm was declared as positive for this pathogen. No information about the kind of production system, production parameters, the severity of clinical signs, the percentage of affected litters, or the number of diseased piglets per litter was collected.

Each fecal content was homogenized, and a swab with Amies medium was taken from this sample. The tube (for PCR testing) and the swab (for bacteriological examination) were sent under cooled conditions to IVD GmbH (Innovative Veterinary Diagnostics Laboratory, Albert-Einstein-Straße 5, 30926 Seelze-Letter, Germany) for analyses.

Bacteriological examination was directly performed after arrival. The feces from the swabs were streaked out on Columbia blood agar (OXOID) and Gassner agar (OXOID) plates (aerobe cultivation), furthermore on Schaedler agar (OXOID) (anaerobe cultivation) and *Clostridium difficile* selective agar (CCFA; OXOID) (anaerobe cultivation, special pre-treatment for *C. difficile*), and cultivated at 37 °C for 24–48 h. Isolates suspected to be either *E. coli*, *C. perfringens*, or *C. difficile* were identified by the typical colony and growth characteristics and confirmed by conventional PCRs for species specific genes. *E. coli* isolates were identified by an in-house-developed PCR using the *E. coli* specific *gapA* gene. The identification of *C. perfringens* was based on the presence of the *cpa*-gene. The PCR was performed according to Baums et al. 2004 [12]. For the identification of *C. difficile*, the PCR described by Doosti and Mokhtari-Farsani in 2014 [13] was used.

Up to three randomly picked *E. coli* colonies of each of the 205 submissions were genotyped individually by an in-house multiplex-PCR developed at IVD GmbH (unpublished) (*n* = 510 colonies). The used multiplex PCRs can detect the virulence-associated genes listed in Table 1 [14]. Based on the combination of detected genes, the *E. coli* isolates could be allocated to four different pathotypes if they encoded for minimum one fimbria/adhesin in combination with one toxin/virulence factor [15] (Table 2).

Up to five *C. perfringens* colonies per submission were analyzed by PCR for genes expressing the alpha- (cpa), beta1- (cpb), beta2- (cpb2), entero- (cpe), epsilon- (etx), and iota-toxin (iap) [12]. These colonies were also examined for alpha- and beta2-toxin production by immunoblot [16].

Up to three *C. difficile* colonies per submission were analyzed by PCR for genes expressing enterotoxin (toxin A), cytotoxin (toxin B), binary toxin A, and binary toxin B [13].

For the virological examination, up to three samples per farm were pooled. Extracted nucleic acids were analyzed by PCR for Rotavirus type A (RVA) [17,18] and by an in-house developed multiplex PCR (unpublished) for the detection of PEDV and TGEV. The occurrence of pathogens and their combinations was analyzed by Mantel-Haenzel-multigroup test (two-sided).

## 3. Results

A total of 555 fecal samples originating from 205 farms, leading to 2.7 samples on average, were submitted and analyzed. In five farms (2.4%), no causative pathogen could be detected. Out of these, 552 fecal samples (99.5%) were positive for *E. coli*. The vast majority of the detected *E. coli* isolates were further investigated by PCR (*n* = 510), and 79 of the farms (38.5%) were positive for *E. coli* belonging to one of the pathotypes ETEC, NTEC, EPEC, or EDEC, as described in Table 2. *C. difficile* could be detected in 232 (42%) samples, leading to 115 (56.1%) positive farms. Rotavirus type A (RVA) was detected in 32.9% of the samples and 71 (34.6%) of the farms. PEDV was present in 2.1% of the samples and, thus, five (2.4%) the farms were positive, whereas no sample was positive for TGEV.

In 380 (68.5%) of the samples, *C. perfringens* could be isolated, and in 173 farms (84.4%) these were assigned to *C. perfringens* type A, while no sample/farm was positive for *C. perfringens* type C (0.0%). In 30 farms (14.6%) *C. perfringens* type A was the only detected pathogen. An overview on the combinations of pathogens that could be found on the 205 farms is shown in Figure 2.

The differences in the occurrence of the single pathogens, as well as the 16 single combinations from Table 2, are significant (*p* < 0.0001, two-sided, Mantel-Haenzel-multigroup-test).

Pathotyping virulence-associated genes were analyzed for 510 randomly picked *E. coli* strains (max. three/submission). Among these genes, encoding for fimA (85.1%), fimH (79.6%), EAST1 (19.4%), papC (8.2%), STII (7.6%), cnf1 (6.3%), aidA (4.9%), and F4 (4.1%) were the most frequently detected (Figure 3).

Based on the combination of virulence genes, 89 (17.5%) *E. coli* isolates could be allocated to one of the four pathotypes defined in Table 2. The most frequently detected pathotypes were 45 (8.8%) enterotoxigenic *E. coli* (ETEC) on 38 (18.6%) farms and 27 (5.3%) necrotoxigenic *E. coli* (NTEC) on 29 (14.2%) farms. Only one EDEC could be isolated on one farm, and 18 (3.5%) EPEC were identified on 15 (7.3%) farms. However, most of the isolates did not belong to a specific pathotype (82.2%). Approximately 50% of the *E. coli* strains belonging to one of these four pathotypes showed hemolysis on blood agar, while only 2.3% of the strains that did not belong to a specific pathotype were hemolytic. In Figure 4, the *E. coli* pathotypes and combinations found on the 205 farms are demonstrated.

A subset of 294 isolates out of the 380 *C. perfringens* isolates (max. 5/submission) were examined by PCR for the toxin genes. All isolates were positive for the alpha-toxin gene, none for the beta1-toxin gene, and 96.6% were positive for the beta2-toxin gene. No strain was positive for the entero-, epsilon-, or iota-toxin genes. Overall, 173 of the farms (84.4%) were positive for *C. perfringens* type A.

The 294 *C. perfringens* type A isolates were tested for their in-vitro toxin production by immunoblot. The most relevant major toxin, alpha toxin alone, was produced by 25 (8.5%) of the isolates, beta2 toxin by 107 (36.4%) isolates, and the combination of both toxins was seen in 44 (15%) isolates. Overall, 176 (59.9%) of the 294 examined *C. perfringens* type A isolates were able to produce alpha and/or beta2 toxin. This led to 122 (59.2%) farms being positive for at least one isolate having the ability to produce alpha and/or beta2 toxin. The difference in the ability of the *C. perfringens* type A isolates either to produce only one or both toxins shown in Figure 5 was significant (*p* < 0.0001, two-sided, Mantel-Haenzel-multigroup-test).

The 234 samples positive for *C. difficile* 47 isolates were further investigated for presence of toxin genes by PCR. All analyzed isolates were positive for the binary toxin A, toxin A, and binary toxin B, and all except three isolates were positive for the gene expressing toxin B. In 116 farms (56%), *C. difficile* could be detected.

## 4. Discussion

Since ND is still a worldwide issue on many pig-producing farms, often causing significant economic losses, current information on the distribution of infectious bacterial and viral pathogens in ND can be helpful for veterinarians to interpret their own diagnostic results and to decide if diagnostics need to be repeated or expanded if commonly detected pathogens are not found in the first attempt, and to better understand the role of detected pathogens on the tested farm.

In 196 out of 205 farms (94.6%), at least one pathogen was detected. Among these 196 positive farms, 68 farms were positive for only one pathogen and 128 (62.4%) farms for two or more pathogens. While a study from 2006 showed that porcine ND normally is a consequence of a single etiological pathogen [19], the results of the present study are in alignment with more recent reports highlighting the involvement of several pathogens in ND and thus underlining multifactorial nature of the disease [3,4,20]. Albeit various combinations between all detected pathogens were found in the present study, toxin producing *C. perfringens* type A in combination with *C. difficile* were most frequently isolated (*n* = 28 [13.7%] farms). On 18 (8.8%) farms, RVA could be detected, additionally. These combinations of etiological agents have also been reported recently in Spain, where regardless of the health status of the piglets, fecal samples were positive for multiple combinations of pathogens [20].

The toxin producing *C. perfringens* type A isolates detected in 57% of the farms was the most frequently found pathogen in this study. Recent studies from Spain and Poland have shown comparable results [5,20,21]. *C. perfringens* type A belongs to the normal intestinal microbiota of healthy piglets but has also been isolated as the only pathogen from piglets suffering from ND. Overall, it is accepted that *C. perfringens* type A can cause ND but is often part of a multifactorial disease that must be considered when a therapy/prophylaxis is planned [5,22]. Johannsen et al. reported general health disorders after an experimental challenge infection with *C. perfringens* type A as well as after intragastric administration of the alpha toxin alone [23]. Most of the *C. perfringens* type A isolates found in this study were positive for the minor toxin beta2 gene (96.6%) An association between the presence of the toxin and the occurrence of enteric diseases in piglets has been described [24]. The present results shows that *C. perfringens* type A encoding for beta2 toxin gene can often (84.4% of the farms) be detected in cases of ND in Germany and also often produce one or even both toxins in vitro (59.2% of the farms).

In contrast, the role of *C. perfringens* type C as a cause for ND seems to be of minor relevance: it has not been detected in this study at all. This is in line with previous investigations on the occurrence of infectious causes of ND [3,4,20,21]. The low detection rate of *C. perfringens* type C in Germany might be explained by vaccination programs, as many farms are vaccinating the gilts/sows before farrowing against enteritis caused by *C. perfringens* type C, using beta-toxoid-containing vaccines in combination with enterotoxigenic *E. coli* as part of a standard animal health procedure.

*C. difficile* is a well-known cause of enteric diseases in humans, especially post-antibiotic treatment [25]. The role of *C. difficile* in porcine ND is still being discussed because it can be found in samples of diarrheic and non-diarrheic piglets. A recent study could show that the prevalence of *C. difficile* is much higher in samples from diarrheic piglets [7]. A retrospective study including data from 2001 to 2010 considered *C. difficile* an emerging pathogen in porcine ND [26]. *C. difficile* and its toxins can cause edema and an accumulation of neutrophils in the mesocolon. Besides diarrhea, in many cases ascites can be observed, and mortality of suckling piglets can reach up to 50% [6]. However, the exact role and pathogenesis of *C. difficile* as a cause of ND still has to be clarified. In the present study, nearly all isolated *C. difficile* strains were positive for the four main toxin genes (93%), which is comparable to the results (86.9%) of the aforementioned study by Kim et al. [7]. In total, 56% of the farms in this study were positive for *C. difficile*. Based on this high detection rate, extended investigations, such as infection experiments, to define the role of *C. difficile* and its virulence factors regarding porcine ND, might be helpful. As foodborne transmission to humans is discussed [5], control programs to get deeper insight on prevalence of the pathogen and implement possible control strategies should be taken into account.

*E. coli*, especially the pathotype ETEC, is well described as primary pathogen and a main reason for porcine ND [27]. Nevertheless, their leading role as the major cause of ND has been questioned in a former study [26]. In the present study, 17.5% of the fecal samples were positive for pathogenic *E. coli* (ETEC, EPEC, EDEC, NTEC). In 6% of the tested farms, they were the only pathogen detected, and in 38% of the farms they were found together with other pathogens. In a recent study from Spain, it was implicated that the presence of pathogenic *E. coli* might have decreased because they were only found in 11% of the investigated farms [4]. Recent data from Germany (obtained during 2015 and 2016) have shown that pathogenic *E. coli* were present in 31% of the pig producing farms with ND [4]. Comparing these results with those of the present study, pathogenic *E. coli* are still one of the most frequently detected pathogens in cases of porcine ND in Germany. Most of the pathogenic *E. coli* strains were typed as ETEC, and among these F4 fimbriae positive ETEC dominated. ETEC positive for F18 fimbriae were also detected, but their role in ND seems to be of no importance, as F18 receptors are expressed on the piglet’s enterocytes not before the third week of life [28]. Due to lack of the receptors on neonate piglet enterocytes, an attaching, release of toxins, and subsequent pathological change cannot happen. Among *E. coli* toxin genes, EAST-1 was most frequently detected. This result differs from the findings of the Spanish study [4], where no isolate encoded positive for EAST-1. These discrepancies show that regional differences in expression of *E. coli* virulence factor genes exist. EAST-1 expressing *E. coli* strains could be found in association with diarrhea in humans, calves, and piglets [29].

NTEC (5.3%) were almost as frequently detected as ETEC (8.8%). The role of this pathotype in porcine ND is still not completely elucidated, but those strains have been described as a cause of ND in humans and ruminants [8,9]. Therefore, further attention should be paid to these pathotypes in future studies. Approximately 82% of the isolated *E. coli* strains could not be allocated to one of the four mentioned pathotypes. The pathotyping, based on a multiplex PCR detecting a selected panel of virulence associated genes, is based on the actual status of knowledge on *E. coli* causing ND [8,9]. As 82.2% of the detected isolates could not be assigned to a pathotype by this method, it cannot be excluded that some of these strains are able to induce ND. These strains are still of great importance for the farms, as increasing numbers of multi-resistant isolates detected in samples of ND in Austria [30] emphasize the importance of a close look on the bacteria involved in ND to decide for the right treatment.

RVA could be detected in 35% of the farms in this study. This shows a slight increase in comparison to previous results obtained from German farms in 2015 and 2016 (27% farms positive for RVA) [3]. In Spain (36% of the tested farms) and South Korea (38% of tested farms) comparable detection rates for RVA could be observed [4,31]. A further study representing data from Spain in the years 2017 and 2018 could show 81% of 31 tested farms with ND being positive for RVA [21]. While occurrence of RVA seems to be comparable in different countries or regions, there is huge diversity among strains belonging to RVA, which could explain different severity in clinical manifestation in piglets suffering from ND [31,32].

Only 2.1% of the farms were positive for PEDV. Since 2014, PEDV outbreaks could be observed in European countries [33] and in Germany [34] but were still only sporadic in Germany. Currently, the detection rates for PEDV seem to be constant [3]. No farm was positive for TGEV. Low or no detection could be expected because outbreaks have been rare over the past years also in other regions in Europe [4].

In 11 farms (5.4%) of this study, no pathogen or potentially pathogenic causative was detectable. Different reasons might have led to this result, such as inadequate transport of samples to the laboratory (no cooling), samples from antibiotic treated piglets, or a non-representative (too low) number of submitted samples per farm [4]. Next, for possible pathogens, numerous non-infectious factors can cause or have an impact on porcine ND. Low environmental temperatures [35], insufficient colostrum intake [36] or age, and condition and feed of the sows [37] are some examples for possible noninfectious factors pushing ND.

In cases of failing to detect a causative pathogen in a first approach, a second submission with a higher number of samples could be helpful to confirm the first result. If potentially pathogenic causatives are not detectable in a second submission, non-infectious reasons could be more likely and should be clarified by further investigations.

Since fecal samples may not reflect the situation in the diseased part of the enteric system, especially in cases multiple infections detected on farms, further investigations such as necropsies in combination with histological analyses can help to find the right solution for the farm.

## 5. Conclusions

This study shows that *C. perfringens* type A, pathogenic *E. coli* such as ETEC, and, especially, F4 fimbriae positive ETEC and RVA are the main infectious agents detected in fecal samples of piglets suffering from ND in Germany. *C. difficile* and NTEC were frequently isolated in diseased piglets, but their role in the pathogenesis of ND remains not completely understood. Further investigations/experiments would be needed to assess their relevance. Many *E. coli* strains could not be allocated to the well-known *E. coli* pathotypes relevant in ND. It would be of interest to look for new virulence factor combinations in *E. coli* strains, which might be involved in ND. Most of the submissions were positive for more than one pathogen, which underlines that ND is probably more often caused by various pathogens. This should be taken into account when a therapy and/or prophylaxis is planned. Compared to other countries the prevalence of pathogens was found to be similar for *C. perfringens* type A and type C as well as Rotavirus type A. Occurrence of pathogenic *E. coli* and their virulence factors are different when comparing results from various countries.

## Figures and Tables

**Figure 1 vetsci-09-00044-f001:**
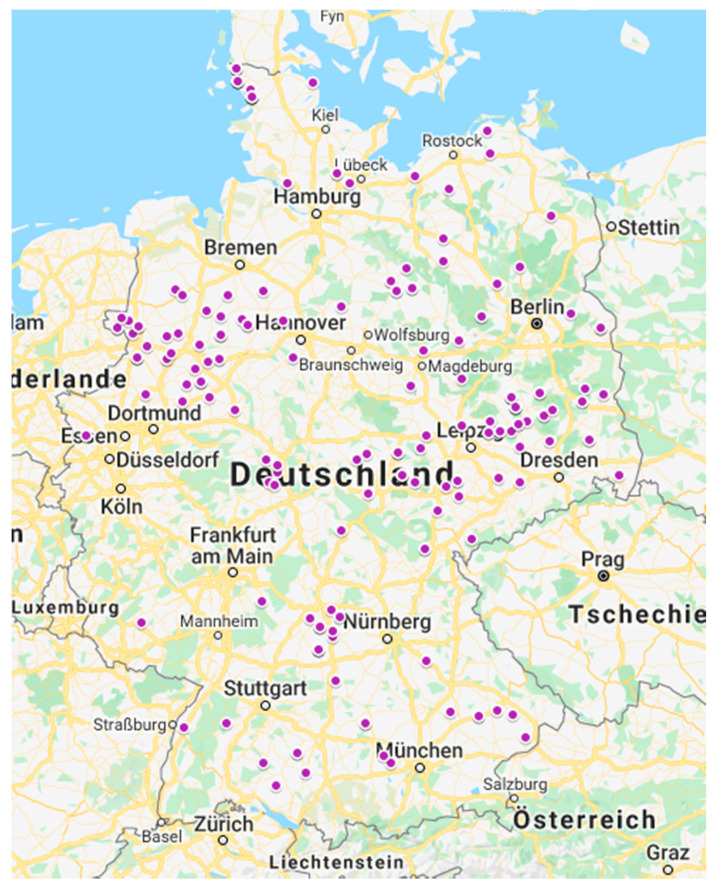
Location of farms participating in the ND study in Germany in 2017.

**Figure 2 vetsci-09-00044-f002:**
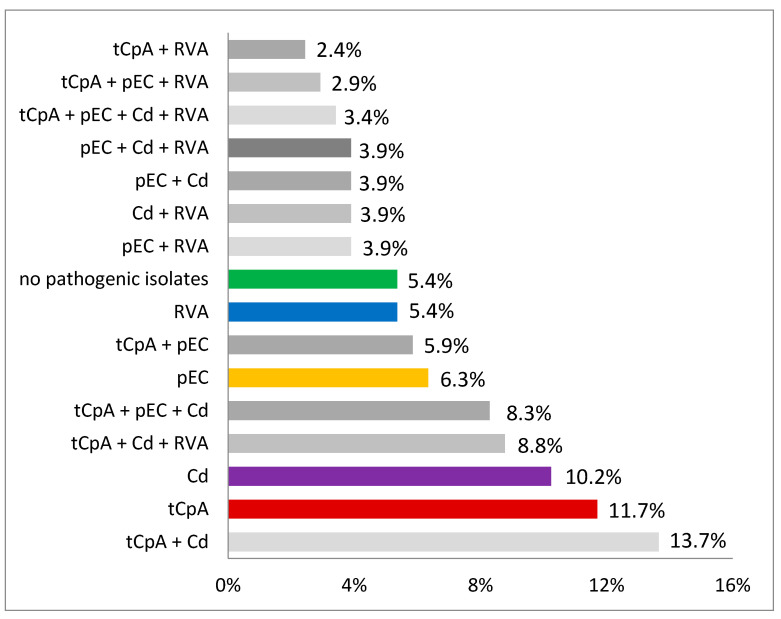
Combinations of pathogens detected in neonatal diarrhea samples in German herds in 2017 (*n* = 205 farms). tCpA: toxin producing *C. perfringens* type A; pEC: pathogenic *E. coli* (ETEC, EPEC, EDEC, NTEC); RVA: Rotavirus type A; Cd: *C. difficile*.

**Figure 3 vetsci-09-00044-f003:**
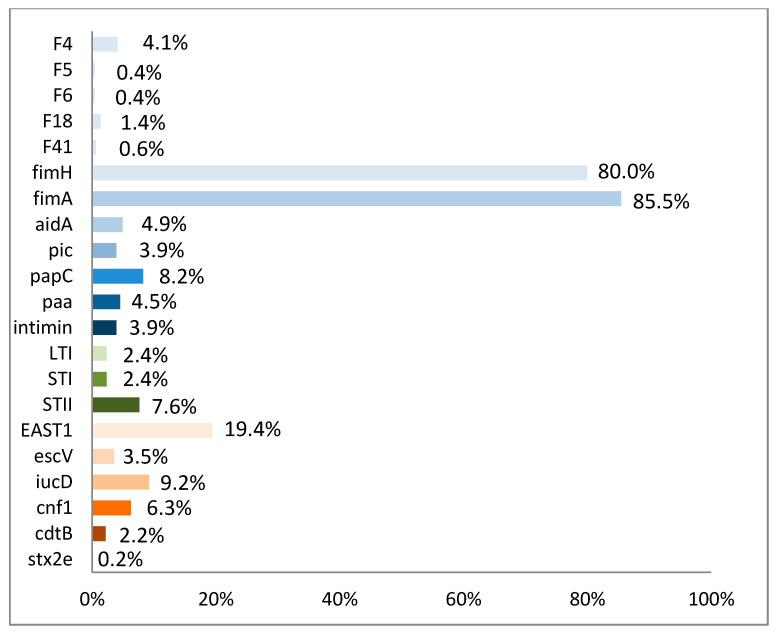
Frequency of detection of virulence factors, adhesins, and toxins detected in *E. coli* isolates (*n* = 510) of samples (*n* = 552) from neonatal diarrhea.

**Figure 4 vetsci-09-00044-f004:**
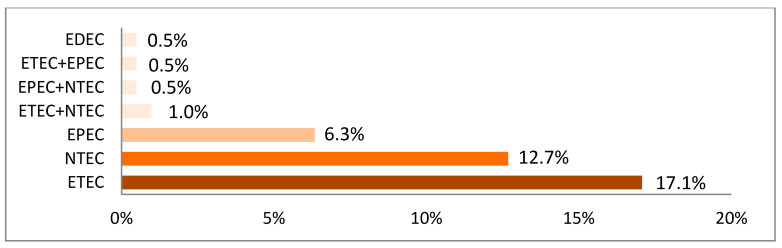
*E. coli* pathotypes and combinations found in neonatal diarrhea samples of 205 farms in Germany (ETEC: enterotoxigenic *E. coli*; EPEC: enteropathogenic *E. coli*; EDEC: edema disease *E. coli*; NTEC: necrotoxigenic *E. coli*).

**Figure 5 vetsci-09-00044-f005:**
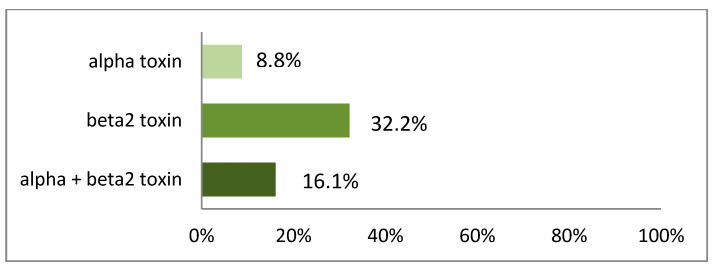
Percentage of farms with at least one *C. perfringens* type A isolate producing alpha-and/or beta2 toxin under in-vitro conditions (immunoblot) in neonatal diarrhea samples of 205 farms in Germany 2017.

**Table 1 vetsci-09-00044-t001:** Target genes of *E. coli* virulence factors detected by multiplex PCRs.

Bacterial Mponents/Products.	Target Genes	Virulence Factors
Fimbriae	*faeG* (F4)	F4 fimbriae
	*faeC* (F5)	F5 fimbriae
	*fasA* (F6)	F6 fimbriae
	*fedA* (F18)	F18 fimbriae
	*fim41A* (F41)	F41 fimbriae
	*fimH* (F1)	Type 1 fimbriae
	*fimA* (F1)	Type 1 fimbriae
Adhesins	*papC*	P fimbriae
	*aidA* (AIDA)	AIDA-I autotransporter adhesin
	*paa*	porcine attaching-effacing associated protein
	*eaeA* (intimin)	intimin
Toxins	*eltB* (LTI)	heat-sensitive enterotoxin
	*estA* (STI)	heat-resistant enterotoxin
	*estB* (STII)	heat-resistant enterotoxin
	*astA* (EAST-1)	heat-resistant enterotoxin
	*stx2e*	Shiga toxin-2e
	*cdtB*	cytolethal distending toxin
Others	*cnf1*	cytotoxic necrotizing factor type 1
	*iucD*	aerobactin siderophore
	*escV*	type III secretion system
	*pic*	serin protease autotransporter

**Table 2 vetsci-09-00044-t002:** *E. coli* pathotypes and their corresponding fimbriae, adhesins, and toxins.

Pathotype	Fimbriae/Adhesins	Toxins/Virulence Factors
ETEC *	F4, F5, F6, F18, F41, AIDA, (paa)	STI, STII, LTI, EAST-1
EPEC *	Intimin, (paa)	escV, (pic)
EDEC *	F18, (AIDA)	stx2e
NTEC *	papC	cnf1, (cdtB)

* ETEC: enterotoxigenic *E. coli*; EPEC: enteropathogenic *E. coli*; EDEC: edema disease *E. coli*; NTEC: necrotoxigenic *E. coli.*

## Data Availability

Data are available from corresponding author upon request.

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
