# Peer review of "Pathogens Detected in 205 German Farms with Porcine Neonatal Diarrhea in 2017"

_vetsci, 2022, doi:10.3390/vetsci9020044_

Round 1

Reviewer 1 Report

The article comprehensively investigates pathogens associated with neonatal diarrhea (ND) in piglets on German farms in 2017. The sample size (555 litters of 205 ND affected farms) can be considered representative and sufficient for descriptive statistics and is comparable to what has been conducted in similar studies in other European countries.

The case definition is clear and material and method part well described. Typing of the investigated bacterial pathogens (Clostridium ssp.; Escherichia ssp.) and detection of virulence factors was performed well, and data are presented in an appropriate way.

The authors highlight the importance of RVA and increasing importance of RVB and RVC as causative agent of ND. Qualitative detection of RVA was performed in the study. Serotyping of RVA and investigations of RVB and RVC in the samples would have been desirable.

The article is well written and of interest for scientific community.

Corrections and suggestions of wording:

Line 59: “ofND” → “of ND”

Line 60f: “In January until December 2017…” → “From January until December 2017…”

Figure 1. “Location of farms attending the ND study in Germany 2017” → “Location of farms attending the ND study in Germany in 2017”

Line 179: “While a studies from 2006 showed that porcine ND” → “While a study from 2006 showed that porcine ND”

Line 243: “should be paid on these pathotypes” → “should be paid to these pathotypes”

Line 248ff: sense to be clarified

Line 259: “RV A” → “RVA”

Author Response

Line 59: “ofND” → “of ND”:

Answer: This has been corrected.

Line 60f: “In January until December 2017…” → “From January until December 2017…”

Answer: This has been corrected.

Figure 1. “Location of farms attending the ND study in Germany 2017” → “Location of farms attending the ND study in Germany in 2017”:

Answer: This has been corrected.

Line 179: “While a studies from 2006 showed that porcine ND” → “While a study from 2006 showed that porcine ND”

Answer: This has been corrected.

Line 243: “should be paid on these pathotypes” → “should be paid to these pathotypes”

Answer: This has been corrected.

Line 248ff: sense to be clarified

Answer: This section has been rephrased. It now reads:

ETEC positive for F18 fimbriae were also detected, but their role in ND seems to be of no importance, as F18 receptors are expressed on the piglet’s enterocytes not before the third week of life [28]. Due to lack of the receptors on neonate piglet enterocytes, an attaching, release of toxins and subsequent pathological change cannot happen.

Line 259: “RV A” → “RVA”

Answer: This has been corrected.

Reviewer 2 Report

Quite interesting study with focus on a specific country. Please find comments in the attached pdf file

Author Response

Line 28: ..other factors ex. Coccidia

Answer: As mentioned in the following sentence, coccidia are rarely detected before the first week of live, and therefore are not seen as a major cause of ND [2].

Line 33: what about Clostridium perfringens Type C…

Answer: C. perfringens Type C is the causative agent of highly fatal, necrohemorrhagic enteritis in newborn piglets and the reviewer is right to mention the bacterium in this context. However, recent studies have demonstrated that the role of C. perfringens Type C decreased, when compared to the situation 20 years ago. Overall, the pathogen plays a minor role in ND [3,4].

Line 65: Independent to quantification, detection levels observed? Was there any connection with the intensity of clinical findings?

Answer: The severity of clinical signs was not recorded by the veterinarians. The samples of clinically diseased litters had to be shipped to the diagnostic laboratory without any information on the percentage of diseased litters or the severity of disease. We added this information in the respective section of the manuscript: No information about the kind of production system, production parameters, the severity of clinical signs, the percentage of affected litters or the number of diseased piglets per litter or production parameters was collected.

Line 66: Was there any connection with the production parameters, alterations, morbidity, mortality, etc.

Answer: No. The authors like to refer to the previous answer.

Line 79: Please add details on PCR methodology used

Answer: The following was added: Isolates suspected to be either E. coli, C. perfringens or C. difficile were identified by typ-ical colony and growth characteristics and confirmed by conventional PCRs for species specific genes.  E. coli isolates were identified by an in-house developed PCR using the E. coli specific gapA gene. The identification of C. perfringens was based on the presence of the cpa-gene. The PCR was performed according to Baums et al. 2004 [14]. For the identification of C.  difficile the PCR described by Doosti and Mokhtari-Farsani 2014 [16] was used.

Line 107+112: Better not start a sentence with a number

Answer: The results section has been rephrased due to the comments of reviewer 3 and all sentences starting with a number were changed as proposed by reviewer 2.

Line 114: ??

Answer: this has been deleted.

Line 144: abbreviations should be explained in every table as self-explanatory

Answer: An explanation of the abbreviations has been added.

Line: 137: please use veterinarians

Answer: This has been changed.

Reviewer 3 Report

Review of “Pathogens detected in 205 German farms with porcine neonatal diarrhea in 2017 by N. Mertens et al.”

Major Comments:

            In this manuscript, Mertens et al. report detection and prevalence of selected primary viral or bacterial pathogens that can cause porcine neonatal diarrhea (ND) in Germany in 2017. A total of 555 samples, each of which is a pool of 5 piglets’ feces, were acquired from 205 farms (Questions: How many samples were collected on average from each farm?; and Have the farms tested similar production system, for example, concrete vs. dirt floor? Please address the answers in text). The fecal samples were tested by RT-PCR for presence of rotavirus A, PEDV, or TGEV and by bacterial cultures for presence of E. coli, C. perfringens, or C. difficile. Isolated bacterial colonies were further tested by PCR or immunoblot for detection of specific toxin or virulence genes or toxins (for C. perfringens type A), respectively. In general, this study shows that at farm level, C. perfringens type A and C. difficile are the most commonly detected pathogens, although C. perfringens type A is known to be a normal gut flora, and to a lesser extent, pathogenic E. coli or rotavirus A - please also attempt to address other distinct trends or findings, for example, the high frequency of co-infections with C. perfringens type A, C. difficile, and rotavirus A as seen in Figure 2.

Overall, this study contains some critical information on a relatively contemporary prevalence of ND causative agents. However, this manuscript seems not clearly written to reveal it. A lot of numbers or percentages within the Result and Discussion sections of this manuscript naturally hinder an easy following and comprehensive understanding of the key data or points. It really takes some time to keep repeating to check the data between the Result and Discussion sections if they are correct and consistent. Therefore, some better ways to describe consistent same numbers or percentages between the two sections, or between at the sample or farm level, need to be created so that some key points can be easily highlighted and taken. For example (but for all text), the sentences that contain numbers or percentages can be more clearly rewritten by showing both farm- and sample-level positivity, as follows: “Alpha toxin gene-positive C. perfringens type A isolates were detected in 173 of 205 farms tested (84.4%) and 380 of 555 samples tested (68.5%). By immunoblot, alpha or beta2 toxin-producing C. perfringens type A isolates were detected in ?? of ?? farms (?%) and ?? of ?? samples (?%). Only alpha toxin-producing C. perfringens type A isolates were detected in ?? of ?? farms (?%) and ?? of ?? samples (?%)”. Only beta2 toxin-producing C. perfringens type A isolates were detected in ?? of ?? farms (?%) and ?? of ?? samples (?%)”. Both alpha and beta2 toxin-producing C. perfringens type A isolates were detected in ?? of ?? farms (?%) and ?? of ?? samples (?%)”.” Particularly, the prevalence of C. perfringens type A based on presence of PCR- or immunoblot-positive alpha toxin should be clearer in Abstract and throughout text.

Minor comments:

  1. Line 17: “122/205 farms (59.2%)” instead of “(57%)”as seen in line 154.
  2. Line 59: “205” instead of “207”.
  3. Line 107 and thorough text: “2.1%” instead of “2,1%”. Please correct all the typos throughout text or in figures or tables.
  4. Lines 115-116: “Figure 2” instead of “table 2”?; and it is quite hard to follow what the differences are, so please refine the sentence so that the readers can easily understand.
  5. Line 149: 173 farms are positive for alpha toxin gene of perfringens type A? Please make the sentence clearer.
  6. Lines 151-152: “under in vitro conditions” means “immunoblot” as seen in Figure 5?
  7. Lines 151-155: The sentences can be confusing to some readers, because the numbers in text are not matching those in Figure 5. For easier understanding of the readers, please mainly describe the findings seen in Figure 5. This suggestion is applied for other text areas along with Figures.
  8. Lines 155-157: “Figure 5” instead of “table 5”?; it is quite hard to follow what the difference is, so please refine the sentence so that the readers can easily understand.
  9. Line 191: “(97%).” instead of “(96%)” based on the data described in line 147.
  10. In Figure 3: “n=552” instead of “n=510” based on the data described in line 104? Please check it.

Author Response

Major Comments:

In this manuscript, Mertens et al. report detection and prevalence of selected primary viral or bacterial pathogens that can cause porcine neonatal diarrhea (ND) in Germany in 2017. A total of 555 samples, each of which is a pool of 5 piglets’ feces, were acquired from 205 farms (Questions: How many samples were collected on average from each farm?; and Have the farms tested similar production system, for example, concrete vs. dirt floor? Please address the answers in text). The fecal samples were tested by RT-PCR for presence of rotavirus A, PEDV, or TGEV and by bacterial cultures for presence of E. coli, C. perfringens, or C. difficile. Isolated bacterial colonies were further tested by PCR or immunoblot for detection of specific toxin or virulence genes or toxins (for C. perfringens type A), respectively. In general, this study shows that at farm level, C. perfringens type A and C. difficile are the most commonly detected pathogens, although C. perfringens type A is known to be a normal gut flora, and to a lesser extent, pathogenic E. coli or rotavirus A - please also attempt to address other distinct trends or findings, for example, the high frequency of co-infections with C. perfringens type A, C. difficile, and rotavirus A as seen in Figure 2.

Answer: Information about the sampling method was added. The section now reads: From each affected litter feces of 5 piglets showing diarrhea, were pooled in a plastic tube. Only feces directly at defecation had to be used, no samples were collected from the floor. Included litters had to be untreated (no antibiotic pretreatment) and between 1 and 7 days of age. If one sample out of a submission was positive for a pathogen the whole farm was declared as positive for this pathogen. No information about the kind of production system, production parameters, the severity of clinical signs, the percentage of affected litters or the number of diseased piglets per litter or production parameters was collected.

As proposed by the reviewer, co-infections found in the present study were discussed in more detail. The section now reads: In 196 out of 205 farms (94.6%) at least one pathogen was detected. Among these 196 positive farms, 68 farms were positive for only one pathogen and 128 (62.4%) farms for two or more pathogens. While a study from 2006 showed that porcine ND normally is a consequence of a single etiological pathogen [19], the results of the pre-sent study are in alignment with more recent reports highlighting the involvement of several pathogens in ND and thus underlining multifactorial nature of the disease [3, 4, 20]. Albeit various combinations between all detected pathogens were found in the present study, toxin producing C. perfringens Type A in combination with C. difficile were most frequently isolated (n=28 [13.7%] farms). On 18 (8.8%) farms RVA could be detected, additionally. These combinations of etiological agents has also been reported recently in Spain, where regardless of the health status of the piglets, fecal samples were positive for multiple combinations of pathogens [23].

Overall, this study contains some critical information on a relatively contemporary prevalence of ND causative agents. However, this manuscript seems not clearly written to reveal it. A lot of numbers or percentages within the Result and Discussion sections of this manuscript naturally hinder an easy following and comprehensive understanding of the key data or points. It really takes some time to keep repeating to check the data between the Result and Discussion sections if they are correct and consistent. Therefore, some better ways to describe consistent same numbers or percentages between the two sections, or between at the sample or farm level, need to be created so that some key points can be easily highlighted and taken. For example (but for all text), the sentences that contain numbers or percentages can be more clearly rewritten by showing both farm- and sample-level positivity, as follows: “Alpha toxin gene-positive C. perfringens type A isolates were detected in 173 of 205 farms tested (84.4%) and 380 of 555 samples tested (68.5%). By immunoblot, alpha or beta2 toxin-producing C. perfringens type A isolates were detected in ?? of ?? farms (?%) and ?? of ?? samples (?%). Only alpha toxin-producing C. perfringens type A isolates were detected in ?? of ?? farms (?%) and ?? of ?? samples (?%)”. Only beta2 toxin-producing C. perfringens type A isolates were detected in ?? of ?? farms (?%) and ?? of ?? samples (?%)”. Both alpha and beta2 toxin-producing C. perfringens type A isolates were detected in ?? of ?? farms (?%) and ?? of ?? samples (?%)”.” Particularly, the prevalence of C. perfringens type A based on presence of PCR- or immunoblot-positive alpha toxin should be clearer in Abstract and throughout text.

Answer: As proposed by the reviewer the result section was rephrased accordingly.

Minor comments:

  1. Line 17: “122/205 farms (59.2%)” instead of “(57%)”as seen in line 154

Answer: This has been corrected.

  1. Line 59: “205” instead of “207”.

Answer: This has been corrected.

  1. Line 107 and thorough text: “2.1%” instead of “2,1%”. Please correct all the typos throughout text or in figures or tables.

Answer: This has been corrected.

  1. Lines 115-116: “Figure 2” instead of “table 2”?; and it is quite hard to follow what the differences are, so please refine the sentence so that the readers can easily understand.

Answer: The whole section has been adapted (refer to ‘major comments’).

  1. Line 149: 173 farms are positive for alpha toxin gene of perfringens type A? Please make the sentence clearer.
  1. Answer: The whole section has been adapted (refer to ‘major comments’).
    1. Lines 151-152: “under in vitro conditions” means “immunoblot” as seen in Figure 5?

Answer: The whole section has been adapted (refer to ‘major comments’).

  1. Lines 151-155: The sentences can be confusing to some readers, because the numbers in text are not matching those in Figure 5. For easier understanding of the readers, please mainly describe the findings seen in Figure 5. This suggestion is applied for other text areas along with Figures.

Answer: The whole section has been adapted (refer to ‘major comments’).

  1. Lines 155-157: “Figure 5” instead of “table 5”?; it is quite hard to follow what the difference is, so please refine the sentence so that the readers can easily understand.

Answer: The sentence has been rephrased. It now reads: The difference in the ability of the C. perfringens type A isolates either to produce only one or both toxins shown in figure 5 was significant (p < 0.0001, two-sided, Mantel-Haenzel-multigroup-test).  

  1. Line 191: “(97%).” instead of “(96%)” based on the data described in line 147.

Answer: This has been changed.

  1. In Figure 3: “n=552” instead of “n=510” based on the data described in line 104? Please check it.

Answer: From the 552 isolated E. coli only 510 were further investigated (max. 3/farm).

Round 2

Reviewer 3 Report

The authors responded appropriately to all the comments raised.